# In Vivo Biodistribution of Respirable Solid Lipid Nanoparticles Surface-Decorated with a Mannose-Based Surfactant: A Promising Tool for Pulmonary Tuberculosis Treatment?

**DOI:** 10.3390/nano10030568

**Published:** 2020-03-21

**Authors:** Eleonora Truzzi, Thais Leite Nascimento, Valentina Iannuccelli, Luca Costantino, Eliana Martins Lima, Eliana Leo, Cristina Siligardi, Magdalena Lassinantti Gualtieri, Eleonora Maretti

**Affiliations:** 1Department of Life Sciences, University of Modena and Reggio Emilia, 41125 Modena, Italy; eleonora.truzzi@unimore.it (E.T.); valentina.iannuccelli@unimore.it (V.I.); luca.costantino@unimore.it (L.C.); eliana.leo@unimore.it (E.L.); 2Laboratory of Pharmaceutical Technology, Federal University of Goiás, Goiânia, Goiás 74605-170, Brazil; thais.leite.nascimento@gmail.com (T.L.N.); emlima@ufg.br (E.M.L.); 3Department of Engineering “Enzo Ferrari”, University of Modena and Reggio Emilia, 41125 Modena, Italy; cristina.siligardi@unimore.it (C.S.); magdalena.gualtieri@unimore.it (M.L.G.)

**Keywords:** tuberculosis, in vivo administration, lipid nanoparticles, alveolar macrophages, active targeting

## Abstract

The active targeting to alveolar macrophages (AM) is an attractive strategy to improve the therapeutic efficacy of ‘old’ drugs currently used in clinical practice for the treatment of pulmonary tuberculosis. Previous studies highlighted the ability of respirable solid lipid nanoparticle assemblies (SLNas), loaded with rifampicin (RIF) and functionalized with a novel synthesized mannose-based surfactant (MS), both alone and in a blend with sodium taurocholate, to efficiently target the AM via mannose receptor-mediated mechanism. Here, we present the in vivo biodistribution of these mannosylated SLNas, in comparison with the behavior of both non-functionalized SLNas and bare RIF. SLNas biodistribution was assessed, after intratracheal instillation in mice, by whole-body real-time fluorescence imaging in living animals and RIF quantification in excised organs and plasma. Additionally, SLNas cell uptake was determined by using fluorescence microscopy on AM from bronchoalveolar lavage fluid and alveolar epithelium from lung dissections. Finally, histopathological evaluation was performed on lungs 24 h after administration. SLNas functionalized with MS alone generated the highest retention in lungs associated with a poor spreading in extra-pulmonary regions. This effect could be probably due to a greater AM phagocytosis with respect to SLNas devoid of mannose on their surface. The results obtained pointed out the unique ability of the nanoparticle surface decoration to provide a potential more efficient treatment restricted to the lungs where the primary tuberculosis infection is located.

## 1. Introduction

Priorities to achieve the WHO goal of ending tuberculosis (TB) epidemic by 2030 include new drugs or formulation approaches to simplify and shorter conventional drug regimens [1]. TB is caused by *Mycobacterium tuberculosis* (Mtb) residing and surviving inside alveolar macrophages (AM). After being inhaled, Mtb reaches the alveoli, where it is phagocytosed by AM, a site difficult to be reached effectively by most antibiotics [2,3,4,5,6]. If the primary infection (pulmonary TB) is not completely eradicated, it can progress and disseminate into other organs (miliary TB). It is important to point out that conventional administration routes, such as oral or parenteral, may lead to sub-therapeutic levels of anti-TB drugs at the primary site of infection, due to poor pulmonary distribution of most systemically administered drugs. As a result, drug-resistant strains may appear quickly [7]. Considering that 75–80% of infection cases remain localized in the lungs, the pulmonary route, widely used for local or systemic treatment of other pathologies [8], appears to be the most promising route of drug administration to reach promptly AM, the primary site of Mtb infection [9,10]. Local administration could harbor resistant tubercle bacilli and avoid first-pass metabolism and gastrointestinal degradation, leading to a reduction in dose level and preventing adverse drug reactions. 

With the purpose to confer on drugs alone the properties required for a powder inhalation therapy in terms of respirability and internalization by AM, particulate vehicles have gained extensive attention. Among these, lipid-based particles are composed of lipids generally recognized as safe and devoid of toxicity following pulmonary administration for most of them [11,12]. Thus, these carriers represent a useful approach for the administration of drugs against pulmonary TB, enabling their deposition onto alveolar epithelia and transport into AM after drug emission by dry powder inhaler devices. Concerning the mechanism of AM entry, phagocytosis is the established mechanism for particles in the range of about 1–3 μm, preserved in macrophages infected by Mtb, providing efficient intracellular drug delivery after only 1 h [4,13,14]. In addition, drug biological activity inside infected AM can be reasonably expected owing to the intracellular biodegradation of the lipid matrix [15].

Within this context, we previously developed respirable solid lipid nanoparticle assemblies (SLNas) loaded with rifampicin (RIF), a clinically useful first-line anti-TB drug, for an AM passive targeting [6,16]. Thereafter an AM active targeting to mannose receptors (MR), located on the macrophage membrane and overexpressed in case of Mtb-infection, was obtained by exploiting mannosylated SLNas. To achieve this, SLNas were surface-decorated with a newly synthesized mannosylated derivative acting as both surfactant, required for the nanoparticle production, and functionalized agent for the AM active targeting. The mannosylated SLNas demonstrated their ability to interact with MR on J774 and MH-S macrophage cell lines improving cell internalization ability in comparison with non-functionalized SLNas and bare RIF. Furthermore, the MR-specific binding was negligibly impaired by the protein/lipid corona layer formed around the nanoparticles upon their contact with a commercial substitute of the natural pulmonary surfactant [6,17].

In addition to AM phagocytosis, other pathways should be considered before establishing a targeting to the alveolar region. Indeed, a few in vivo examples of translocation across the alveolar epithelium through the interstitium into the blood or lymphatic circulation were reported [18,19,20,21,22], even if the mechanism involving inhaled nanoparticle alveolar clearance remains quite unknown. It therefore follows that in vivo experimental studies have to be performed to assess the actual fate of inhaled particles, especially for those having new surface physicochemical properties. 

Thus, the purpose of the present work was to detect in mouse lungs and extra-pulmonary organs the biodistribution of the previously developed SLNas surface-decorated with the mannose surfactant in comparison with non-functionalized samples and bare RIF.

## 2. Materials and Methods 

### 2.1. Materials

Palmitic acid (PA) was purchased from Fluka Chemie (Buchs, Switzerland), cholesteryl myristate (CM) from TCI Europe (Zwijndrecht, Belgium), polyoxyethylene–polyoxypropylene block copolymer (Lutrol F127) (F127) from BASF (Ludwigshafen, Germany), sodium taurocholate (ST), rifampicin (RIF) from Alfa Aesar (Ward Hill, MA, USA), IR-780 iodide from Sigma-Aldrich (Milan, Italy), and RPMI 1640 from PAN-Biotech (Aidenbach, Germany). The mannose derivative hexadecanoic acid (aminoethyl α-D-mannopyranoside) amide (MS) was synthesized as previously reported [6] and used as mannosylated functionalizing/surfactant agent. For HPLC analysis, methyl parahydroxybenzoate, used as internal standard (IS), was purchased from Carlo Erba Reagenti (Milan, Italy). For the in vivo experiments, ketamine and xylazine were purchased from Syntec (Santana de Parnaiba, São Paulo, Brazil), hematoxylin–eosin (H&E) from Easypath (Indaiatuba, São Paulo, Brazil), ascorbic acid from Merck (São Paulo, Brazil), Hoechst 33,342 from Invitrogen (Carlsbad, CA, USA), and Tissue-Tek O.C.T. Compound from Sakura (Alphen aan den Rijn, Netherlands). All the other chemicals were of analytical grade. 

### 2.2. SLNas Preparation

SLNas were prepared by the melt emulsification technique, as previously described in detail [17]. Briefly, the lipid phase containing 92.5 mg PA, 42.5 mg CM, and 45 mg RIF was melted and emulsified by ultrasonication (Branson Sonifier^®^ SFX150, Emerson, St. Louis, MO, USA) (input 150 W for 3 min) in 10 mL MilliQ water containing the surfactant/functionalizing agent MS (0.1%) alone or in a blend with ST (0.05%) to prepare SLNas/MS or SLNas/MS-ST samples, respectively. As the controls, non-mannosylated SLNas/ST and SLNas/F127 samples were obtained by using 0.2% ST or 1% F127, respectively, as emulsifier. After cooling in ice bath, SLNas samples were purified by dialysis, frozen at −70 °C, and freeze-dried (Lio 5P, CinquePascal, Milan, Italy) according to the method previously adopted [23]. All the samples were used in a labeled form by dissolving preventively 0.2% IR-780 in the melted PA before the melt emulsification process.

### 2.3. Morphology, Size, and Z-Potential

SLNas morphology was evaluated by Scanning Electron Microscopy (SEM, Nova NanoSEM 450, Fei, Eindhoven, The Netherlands) using TEM mode with STEM detector (30 kV) on SLNas coated with carbon (carbon coater, Balzers CED- 010, Oerlikon Balzers, Liechtestein). Photon Correlation Spectroscopy (PCS) (Zetasizer version 6.12, Malvern Instruments, Worcs, U.K.) equipped with a 4 mW He-Ne laser (633 nm) and a DTS software (Version 5.0) was used to measure size, polydispersity index (PDI), and Z-potential values. Before the test, all the samples were suspended in MilliQ water for Z-potential analyses or RPMI 1640 medium for size and PDI determination, vortexed for 1 min and dispersed for 3 min in ultrasonic bath (USC200TH, VWR International, Milan, Italy) at 37 °C. Each sample was analyzed in triplicate.

### 2.4. HPLC Analysis

Quantification of RIF in both mice organs and SLNas samples was carried out using a 1260 Infinity HPLC (Agilent Technologies, Santa Clara, CA, USA) equipped with a 1260 quaternary pump, 1260 DAD detector, and 1260 ALS flow cell. Chromatographic separation was achieved using a Zorbax Extend C18 column (150 × 4.6 mm, 5 μm) (Agilent Technologies) at 35 °C. The mobile-phase consisted of a mixture of acetonitrile (ACN):water (v/v) with flow rate of 1.0 mL/min and used in a gradient as follows: 30:70 until 2 min, increasing to 70:30 at 2–4 min, kept at 70:30 until 7 min, decreasing to 30:70 from 7–9 min and kept at 30:70 until 11 min. Detection wavelengths were 254 nm and 333 nm (for plasma samples) and the injection volume was 10 μL. For the drug loading determination, a calibration curve from 5 to 30 μg/mL was obtained with r^2^ > 0.995. Calibration curves in each organ were obtained from 0.09 to 2.7 μg/mL and from 3 to 7 μg/mL with r^2^ > 0.988, using IS. Run time was 11 min, and RIF elution was observed at 5.4 min and IS elution at 3.7 min. 

### 2.5. Drug Loading Levels

To 10 mg of SLNas samples, 10 mL of IS solution and methanol were added. The suspension was heated at 50 °C for 30 min, properly diluted (1:5) with methanol, filtered, and analyzed by HPLC. Drug loading (DL%) was calculated as the weight percentage of RIF in the total particle weight. The reported values were averaged on three determinations.

### 2.6. In Vitro Release

In vitro RIF dissolution and release from SLNas powders were examined in sink conditions on exactly weighed samples (100 mg) by means of the dialysis membrane (MWCO 12–14,000 Da; Medicell International Ltd., London, GB) method, in 30 mL of simulated lung fluid (SLF) according to Marques [24] under gently magnetic stirring at the temperature of 37.0 ± 0.2 °C to reproduce stagnant lung conditions. Sample solutions (1 mL) were withdrawn at fixed intervals for a time period of 6 h and the initial volume restored. RIF quantification was performed spectrophotometrically (Lambda 35; Perkin-Elmer, Norwalk, CT, USA) at the wavelength of 475 nm. Sample solutions were also monitored for IR-780 release from SLNas by spectrophotometric quantification at the wavelength of 783 nm. The reported values were averaged on three determinations. 

### 2.7. In Vivo Study

#### 2.7.1. Animals

Female Swiss mice aged 4–6 weeks (27–35 g) were obtained from the Central Animal Facility at Federal University of Goiás (Goiânia, Goiás, Brazil). Considering the preliminary study phase, healthy animals were used. Animals were acclimatized for a week prior to the beginning of experiments under 12:12 h light–dark cycles and controlled temperature. Food and water were provided ad libitum and animals were treated with chlorophyll-free grains for 7 days before treatment. Experiments were conducted according to the NC3RS guidelines for laboratory animal care. The experimental protocol (no. 109/18) was approved by the Animal Research Ethics Committee of the Federal University of Goiás. Animals were anesthetized before sample administration with ketamine and xylazine at 100 and 10 mg/kg, respectively. The powder aerosolization followed the method described by Chaurasiya et al. [25]. Briefly, after intubation, 3 × 350 µL puffs of air were used to administer the powders into the mouse trachea using 20 GA × 1.16″ catheters connected to 1 ml syringe.

#### 2.7.2. Real-Time Fluorescence Imaging

Four groups of mice (n = 4 per group) were treated with 3 mg of SLNas/MS, SLNas/MS-ST, SLNas/ST, and SLNas/F127 formulations. An untreated group was used as the control. Real-time fluorescence imaging was used to evaluate qualitatively the tissue distribution of SLNas formulations. Fluorescence scans were performed at 0.5, 3, 6, and 24 h after administration using a fluorescence molecular tomography (FMT) in vivo imaging system (model 1500, Perkin-Elmer, Norwalk, CT, USA). Excitation and emission quantification wavelengths were 790 nm and 780 nm. Animals were euthanized 24 h after treatment using an excess of anesthetic. Lungs, heart, liver, and kidneys were harvested, rinsed with 0.9% NaCl solution and imaged for fluorescence detection by FMT.

#### 2.7.3. Lung Section Analysis

At 24 h after intratracheal administration of SLNas samples, lungs were dissected and flash-frozen with liquid nitrogen in Tissue-Tek O.C.T. Compound. Transverse sections of 6 μm thickness were obtained at various points along the length of the tissue (superior, median, and caudal regions) using a Leica CM-1850 cryostat (Leica Biosystems, Inc., Buffalo Grove, IL, USA). For histopathological evaluation, the sections were stained with H&E and analyzed using light microscopy (DM 2000 Leica Microsystems, Bannockburn, IL, USA) coupled to a photographic camera (Canon PowerShot S80, VA, USA). Moreover, lung sections were observed by fluorescence microscopy (DMI 4000B, Leica Microsystems, Bannockburn, IL, USA).

#### 2.7.4. Rifampicin Biodistribution

To quantify the amount of RIF in the different tissues after intratracheal administration, mice were randomly divided into five groups (n = 4–6 per group) and treated with 3 mg of SLNas/MS, SLNas/MS-ST, SLNas/ST, and SLNas/F127 or bare RIF diluted with 3 mg of mannitol (10% RIF) to obtain insufflation RIF dose comparable to those of SLNas samples. An untreated group was used as the control. Mice were euthanized 30 min and 3 h after treatment. 

#### 2.7.5. Actual Inhaled Drug Dose

The non-administered SLNas powder was quantified in order to calculate the actual inhaled dose of RIF per mouse. In practice, each administration device was washed 10× with 2 mL of ACN containing IS. The resulted suspension was heated at 50 °C for 30 min and filtered (0.45 μm porosity). The obtained solution was analyzed by HPLC. The inhaled dose, expressed as percentage of the administered one, was calculated by subtracting the non-administered powder to the amount introduced into the administration device. 

#### 2.7.6. Bronchoalveolar Lavage Fluid

Bronchoalveolar lavage fluid (BALF) was collected as described by Aragao-Santiago [26]. Alveolar macrophages were separated from the BALF by centrifugation at 500× *g* for 10 min. The cell pellet was then split in two aliquots for AM fluorescence microscopy and RIF quantification. For fluorescence microscopy, BALF cells were smeared in duplicates onto slides using Cyto-System (Hettich, Germany) at 219 g for 10 min, at room temperature. Cytoslides were then stained with Hoechst 33,342 and observed using fluorescence microscopy. For RIF quantification, cells were lysed by the addition of 200 μL ACN containing IS and sonicated by ultrasonic bath for 15 min. The solutions were filtered (0.45 μm porosity) and analyzed.

#### 2.7.7. Plasma and Organs

Blood was collected by cardiac puncture and immediately centrifuged at 2000× *g* for 5 min to separate the plasma, which was stored overnight at −70 °C before analysis by HPLC. Lungs, liver, and kidneys were harvested, rinsed with 0.9% NaCl, dried, and weighed. Tissues were homogenized in pH 4.5 phosphate-buffered saline (PBS), containing 10^−3^ M of ascorbic acid, by Ultra-Turrax (Ika Labortechnik, Staufen, Germany) at 20,000 rpm for 3 min in glass conical tubes placed in ice. The homogenates were stored at −70 °C overnight prior to HPLC analysis. RIF was extracted from tissues and plasma by protein precipitation. 300 μL of homogenate and 150 μL of plasma were treated with 1 mL and 500 μL of ACN (4 °C), respectively, and mixed by vortex for 60 s. The mixtures were centrifuged at 5000× *g* for 5 min (4 °C). The supernatants were collected, filtered (0.45 μm porosity), and analyzed. The IS, diluted with ACN, was added to all samples to a final concentration of 0.69 μg/mL prior to protein precipitation. 

### 2.8. Statistical Analysis

Statistical analysis of data was performed by two-way analysis of variance (ANOVA) followed by Bonferroni’s test. Differences between groups were considered to be statistically significant at *p* < 0.05.

## 3. Results and Discussion

In our previous works, respirable mannosylated SLNas/MS and SLNas/MS-ST samples were designed to exploit the benefit of an active targeting via mannose receptor-mediated mechanism on AM internalization ability. Mannosylation was achieved by anchoring on SLNas surface a newly synthesized mannose surfactant (MS) alone or in a blend with sodium taurocholate (ST). Indeed, mannose derivatives show selectivity for multiple C-type lectin-like domains on mannose receptor (MR) whereas ST was selected for its absorption enhancing ability as well as a possible recognition of the N-terminal cysteine-rich domain on MR [27]. Non-mannosylated SLNas/ST and SLNas/F127, produced, respectively, by using ST alone and F127 that does not bear groups able to bind MR, were investigated as the controls. The samples were characterized for particle geometrical properties, flowability, physical state, wettability, respirability performance, mannosylation efficacy, and protein/lipid corona formation. Moreover, cytotoxicity, actual MR involvement, and RIF transport within alveolar macrophage MH-S cell line, also in the presence of a pulmonary surfactant substitute, were assessed. The mannosylated samples satisfied the properties required for reaching the alveolar epithelium in terms of respirability parameters. Moreover, the functionalization by means of mannose groups, in particular on SLNas/MS surface, improved RIF translocation into MH-S cells without impairment caused by the corona layer covering the nanoparticles [17].

### 3.1. In Vitro Properties

The present investigation was carried out to establish in vivo RIF biodistribution following intratracheal administration in mice of SLNas/MS and SLNas/MS-ST samples in comparison with the controls SLNas/ST and SLNas/F127. The samples were used in their labeled state by means of IR-780 dye. Consequently, the same batch of each SLNas sample to be administered to mice was analyzed to verify morphology, size, Z-potential, drug loading, and in vitro release in SLF.

All the SLNas samples appeared mainly as aggregates of smaller nanoparticles (Figure 1). These assemblies were rounded in shape with a size of the main class (>85%) in the range from 450 nm to 850 nm and DPI values from 0.41 to 0.71 owing to the presence of minor populations (<15%). Z-potential values of the samples were from −8.5 to −55 mV. The most negative values were due to MS and ST charges whereas the lowest magnitude was conferred by the non-ionic F127 (*p* < 0.05). Rifampicin payload in SLNas was about 10% without relevant differences among the samples (Table 1).

Concerning in vitro release, which allows for predicting drug leaching before AM internalization, RIF release profile from SLNas was gradual compared with the fast and complete free RIF dissolution. About 30% RIF was released from mannosylated SLNas/MS and SLNas/MS-ST at the end of the experiment, less than the values detected from the non-mannosylated samples (Figure 2), probably owing to a stronger barrier offered by the mannosylated surfactant [6]. In the light of the fast internalization process by AM, acceptable drug retaining within SLNas matrices before macrophage uptake would be therefore reasonably expected. IR-780 was not registered in the release medium throughout the experiment.

### 3.2. In Vivo Study

The transport of RIF within pulmonary and extra-pulmonary regions by means of mannosylated SLNas (SLNas/MS and SLNas/MS-ST) was assessed in mice following intratracheal administration of 3 mg dose of powder, i.e., the maximum tolerated dose. The study involved the real-time whole-body imaging on living mice and excised organs (lungs, liver, and kidneys) as well as the fluorescence microscopy of AM from BALF and lung dissections. Additionally, RIF was quantified in AM, lungs, plasma, liver, and kidneys. Liver and kidneys were selected as extra-pulmonary organs for their relevance in RIF systemic pathway, accumulation, and metabolism. Drug transport was compared with that provided by administering non-mannosylated SLNas (SLNas/ST and SLNas/F127) and bare RIF. Short-term studies were performed at quantification times of 0.5 h and 3 h post-exposure since AM internalization as well as pulmonary and extra-pulmonary effects occurs within the first hours after administration [28,29,30].

FMT performed on mice and fluorescence microscopy on AM collected from BALF exploited the fluorescence emitted from IR-780 retained within SLNas, as demonstrated by the in vitro release study. Indeed, the highly hydrophobic nature of this dye allowed its firm non-covalent embedding in several lipid structures [31,32].

Concerning RIF determination, data generating from each animal in terms of drug concentration were related to the drug loading corresponding to each administered sample. Moreover, since relationships between dosage emission and device flow-rate are well known [33,34,35], the actual inhaled dose per mouse was calculated by considering RIF amount not emitted by the device. The actual emission doses were ≥70% of those loaded inside the device (78.6 ± 3.3%, 69.3 ± 9.2%, 85.6 ± 5.4%, for SLNas/MS, SLNas/MS-ST, SLNas/ST, respectively) apart from SLNas/F127 with an emission dose of 53.4 ± 10.7% owing to its very poor flowability and high cohesiveness already highlighted in our previous work [17]. 

#### 3.2.1. Fluorescence Imaging on Mice Whole-Body and Excised Organs

SLNas accumulation and retention within lungs after intratracheal administration was assessed by real-time FMT. The analysis was carried out for 24 h in order to monitor particle permanence in lungs and particle spreading to all body, by measuring IR-780 fluorescence intensity emitted from living mice whole-bodies. Moreover, excised organs (lungs, liver, and kidneys) were analyzed 24 h post-exposure. Taking into account nanoparticle ability to penetrate into rodent lung interstitial spaces [36], SLNas detection in lungs was assigned to both particle translocation across epithelial cells and internalization by AM which were not removed by BALF.

Despite IR-780 strong fluorescence intensity and low fluorescent mice diet used in the present study, mice auto-fluorescence limited the monitoring of SLNas distribution by means of FMT [37]. Nevertheless, increases in IR-780 accumulation compared with the fluorescence from untreated animals were registered only in the lungs of SLNas/MS-treated mice. Concerning mice organs excised at 24 h, fluorescence was registered only in the lungs, regardless of SLNas sample, even though a higher intensity was observed after SLNas/MS administration (Figure 3). This finding may offer a preliminary evidence of a more substantial translocation and long-term accumulation within lungs of the nanoparticles surface-decorated only with MS as compared with the other SLNas samples. Appreciable IR-780 emissions from extra-pulmonary regions were not detected, regardless of SLNas sample, demonstrating the high retention of SLNas in the pulmonary region with a poor diffusion in the circulatory system. 

Finally, no IR-780 signal was found in the upper respiratory region at all scanning times, suggesting the nanoparticle deposition into the lower respiratory tract, without any mucociliary clearance. 

#### 3.2.2. SLNas Biodistribution in the Pulmonary Region

In order to further investigate the localization and permanence of SLNas samples within the pulmonary region, fluorescence microscopy on AM from BALF and lung tissue sections was performed. Subsequently, RIF quantification in both AM and lung tissue was investigated. Once nanoparticles are deposited onto the alveolar region of the respiratory tract, they first come into contact with pulmonary surfactant and then they are taken up by AM mostly within 2–3 h after particle deposition [19]. Hence, BALF was isolated in order to remove non-internalized nanoparticles from the lungs and collect non-adherent AM, i.e., the cells residing in the airway lumen and representing more than 90% of the cells contained in the mouse BALF [38,39,40]. As a further result, non-adherent AM were separated from interstitial macrophages, a large component of the total macrophage population in murine lungs residing in lung parenchyma [41]. AM recovery from BALF, however, has been reported to be incomplete since the most activated AM by particle internalization strongly adhere to the lung tissue [41,42]. The morphologic study of BALF cell population by optical microscopy observation confirmed that the majority of cells were AM. No significant differences in the total number of macrophages collected in BALF were observed among the administered SLNas samples as well as in comparison with that obtained from the untreated mice, regardless of the collection time. Compared with the control (cells from untreated mice), IR-780 particle-related fluorescence was observed in the collected AM for SLNas/MS, SLNas/MS-ST, and SLNas/ST both at 0.5 h and 3 h post-exposure (Figure 4). This finding might already evidence the AM internalization of the mannosylated samples as well as SLNas bearing ST on their surface, nearly in agreement with previous in vitro results [17]. In contrast, negligible fluorescence intensities were detected for SLNas/F127, regardless of the collection time. Cell internalization appears, therefore, to be precluded in the case of SLNas/F127 as support of its insufficient respirability properties [17] and the known cell adhesion inhibition ascribed to F127 chains [43]. 

Lung dissections at 24 h post-exposure observed by fluorescence microscopy indicated that a portion of SLNas was still present in the lung tissue, regardless of SLNas sample (Appendix A), in agreement with FMT results. In the early post-exposure time points (0.5 and 3 h), the relationship between SLNas detected within AM by means of fluorescence microscopy and RIF quantification in the lungs might offer information about nanoparticle entry mechanisms according to their surface properties. The highest RIF levels in lungs (*p* < 0.001) were found following SLNas/MS administration (19.65 ± 10.97% and 2.36 ± 2.65% of the inhaled dose per gram of tissue at 0.5 and 3 h, respectively). Otherwise, RIF levels were about 10-fold lower following 0.5 h exposure to SLNas/MS-ST and SLNas/ST (1.87 ± 1.97% and 2.35 ± 2.41% of the inhaled dose per gram of tissue, respectively) and about 4.5-fold lower for bare RIF (4.37 ± 3.30% of the inhaled dose per gram of tissue) without significant differences among them (*p* > 0.05). Then, RIF levels declined up to the quantification limit or to negligible concentrations at 3 h post-exposure (Figure 5). Only SLNas/MS provided RIF levels higher than the minimum inhibitory concentration against Mtb strain [30] regardless of the post-exposure time. 

The greater RIF retention in the lungs measured at 0.5 h post-exposure to SLNas/MS might have received the major contribution from internalization by adherent AM via the mannose-receptor mediated pathway in accordance with the results obtained previously on murine MH-S cells [17]. Although SLNas/MS-ST and SLNas/ST were detected within AM as well, the lower immediate retention of RIF in lungs compared with that achieved by SLNas/MS administration could be related to their passive mechanism of entry AM without mannose receptor involvement [17]. Despite this, the role of the carrier surfaces modified by MS in a blend with ST as well as ST alone would ensure a certain intramacrophagic activity in contrast to the inhalation of bare RIF. Indeed, drug localization in lungs post-exposure to bare RIF, showing difficulty of crossing AM membrane as demonstrated previously [6], could be reasonably assigned to a translocation mechanism across epithelial cells into the interstitium or other lung cells. Declining RIF concentrations at 3 h post-exposure was observed for all the groups. The decrease of RIF levels could be attributed to both AM clearance involving RIF biodegradation [6] and systemic translocation. Other pathways for lung entrance and subsequent clearance should be, however, considered for SLNas bearing ST on their surface. In fact, in addition to AM internalization, the paracellular passage should be feasible owing to the well-known permeation enhancing properties of ST [44]. Otherwise, drug lung clearance following 3 h post-exposure to bare RIF might be due to the molecule absorption beyond the epithelial barrier into the blood and lymphatic circulation. No RIF amounts, conversely, were measured in the lungs at both 0.5 and 3 h post-exposure to SLNas/F127 in agreement with the lack of AM fluorescence and RIF levels determined previously on MH-S cell line [17]. This finding is consistent with the non-adhering properties of F127 surfactant to cells, preventing nanoparticle translocation across both AM and epithelial cells. No RIF was detected within non-adherent AM collected from the BALF post-exposure to all the samples, regardless of the exposure time probably owing to the quantification limit of the HPLC method.

From a histopathological point of view, lung sections examined at 24 h post-exposure to SLNas/ST, SLNas/MS-ST, and SLNas/F127 samples indicated the absence of an inflammatory response. No signs of toxicity were expected, since all the samples were obtained by using recognized biocompatible excipients [45]. Moreover, in vivo administration of RIF in rats did not exhibited any evidence of toxicity after inhalation [46]. On the contrary, a mild neutrophilic infiltration in mice treated with SLNas/MS was noticed (Figure 6) even if MR activation has been generally recognized as a promoter of the anti-inflammatory response [47,48,49]. Thus, a contribution to the inflammation might arise from the increased SLNas/MS endocytosis compared with the other samples, leading to a higher production of reactive oxygen species [50]. 

#### 3.2.3. SLNas Biodistribution in Extra-Pulmonary Regions

Inhaled nanoparticles have been proved to pass through the lung epithelium, enter the blood circulation, and accumulate in extra-pulmonary organs, mainly according to the nanoparticle size [21]. Translocation from the lungs to other organs with a possible accumulation in lymph nodes may occur from direct uptake into epithelial cells and/or phagocytosis by AM, which then migrate into the lung interstitium [26,51]. Thus, plasma, liver, and kidneys were analyzed by both RIF quantification at 0.5 and 3 h post-exposure to SLNas samples and bare RIF as well as FMT at 0.5, 3, 6, and 24 h post-exposure to SLNas samples. However, regarding the recovery of RIF in plasma, RIF serum half-life in mice (about 5 h) needs to be taken into account [52,53]. In addition, plasma levels are transient since they are balanced with the clearance by the kidneys and uptake by the other tissues. Drug levels in plasma and liver were found less than 2% for all the samples, regardless of the exposure time, without significant differences among them (*p* > 0.05). These findings indicate the poor systemic activity offered by the lipid nanocarrier, irrespective of the pathway followed from the airway lumen into the lung tissue. Similarly, other studies in mice have evidenced negligible levels in blood and extra-pulmonary organs for nanoparticles >30 nm in size [18,21,26]. Drug levels in kidneys were barely measurable, regardless of the sample and the exposure time (Figure 5). 

## 4. Conclusions

The present in vivo study was addressed to assess the fate of rifampicin transported by respirable mannosylated lipid-based nanoparticles designed for an intramacrophagic delivery. The results highlighted the suitability of the RIF-loaded nanocarrier for efficiently targeting the alveolar macrophages via a mannose-receptor mediated pathway with a poor systemic biodistribution. The achievement of antimicrobial drug concentrations at this site of infection may support the goal of a potential application as inhaled therapy for the treatment of the pulmonary tuberculosis.

## Figures and Tables

**Figure 1 nanomaterials-10-00568-f001:**
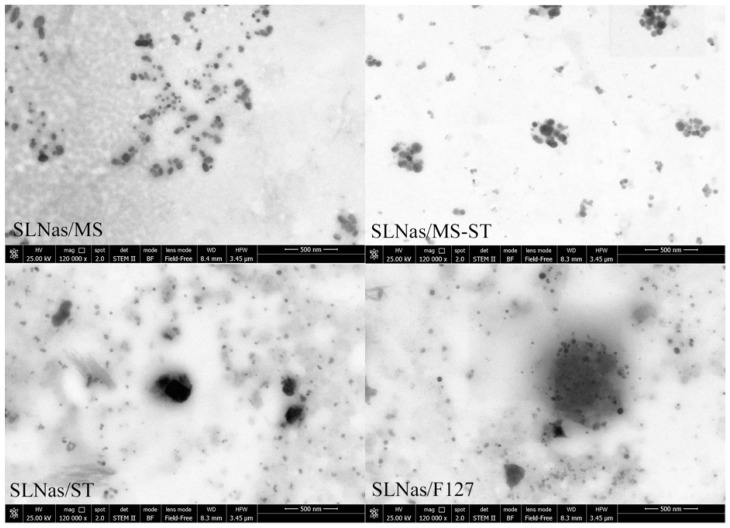
SLNas images by transmission electron microscopy.

**Figure 2 nanomaterials-10-00568-f002:**
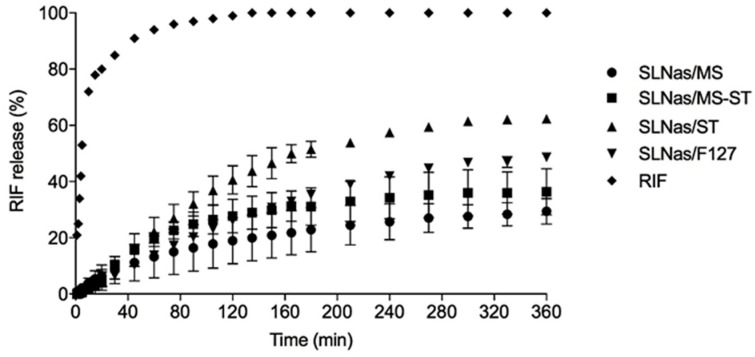
In vitro rifampicin (RIF) dissolution and release from SLNas samples in simulated lung fluid. For some points error bars do not exceed symbol size.

**Figure 3 nanomaterials-10-00568-f003:**
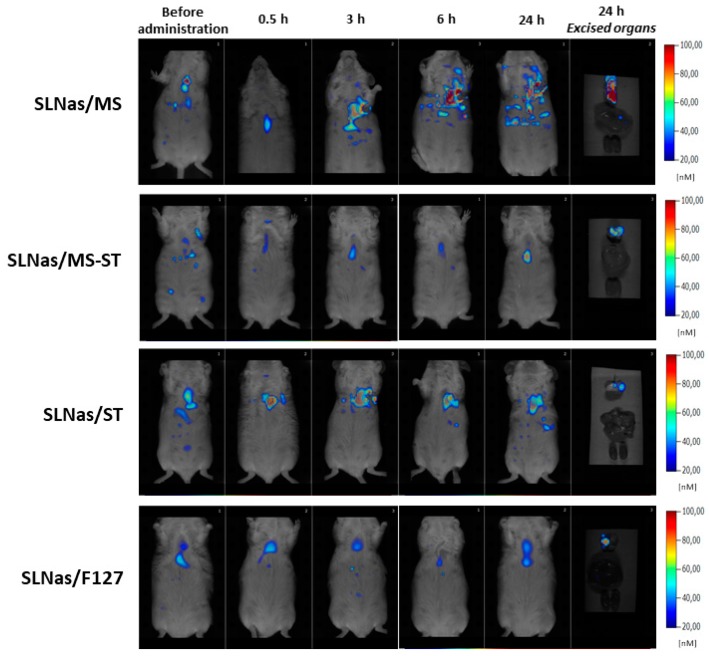
Representative FMT images at 0.5, 3, 6, and 24 h after administration of SLNas to mice. Whole-body scans were acquired after pulmonary treatment of SLNas/MS, SLNas/MS-ST, SLNas/ST, and SLNas/F127 samples. Images of the excised organs (lungs, liver, and kidneys from top down) show the organ biodistribution of particle-related fluorescence at 24 h post-exposure.

**Figure 4 nanomaterials-10-00568-f004:**
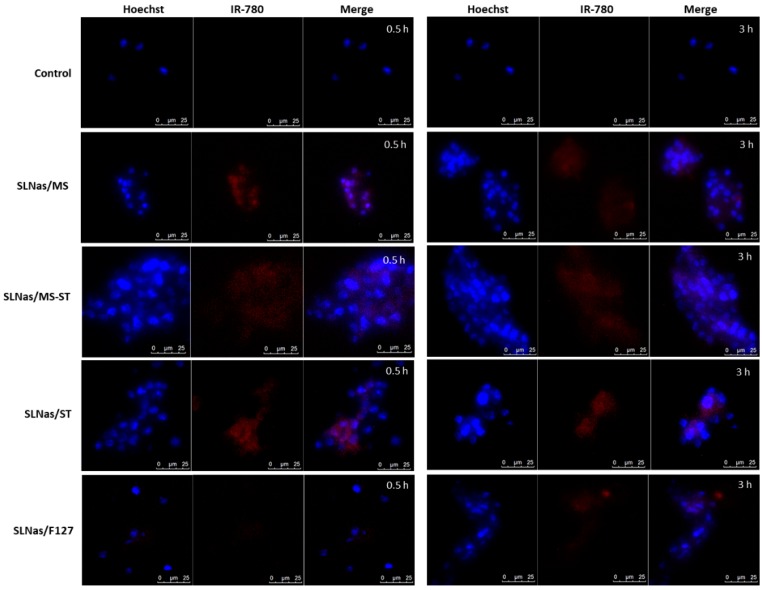
Fluorescence microscopy on AM from BALF showing SLNas post-exposure. Pictures were taken with a 40× magnification. From left to right, first frame: image taken using the filter for Hoechst staining (cells); second frame: image taken using the filter for IR-780 (SLNas); third frame: merge, showing BALF AM in blue and IR-780/SLNas in red. Animals were euthanized after 0.5 (right panel) and 3 h (left panel) post-exposure and BALF was immediately collected for analysis. Scale bars = 25 µm.

**Figure 5 nanomaterials-10-00568-f005:**
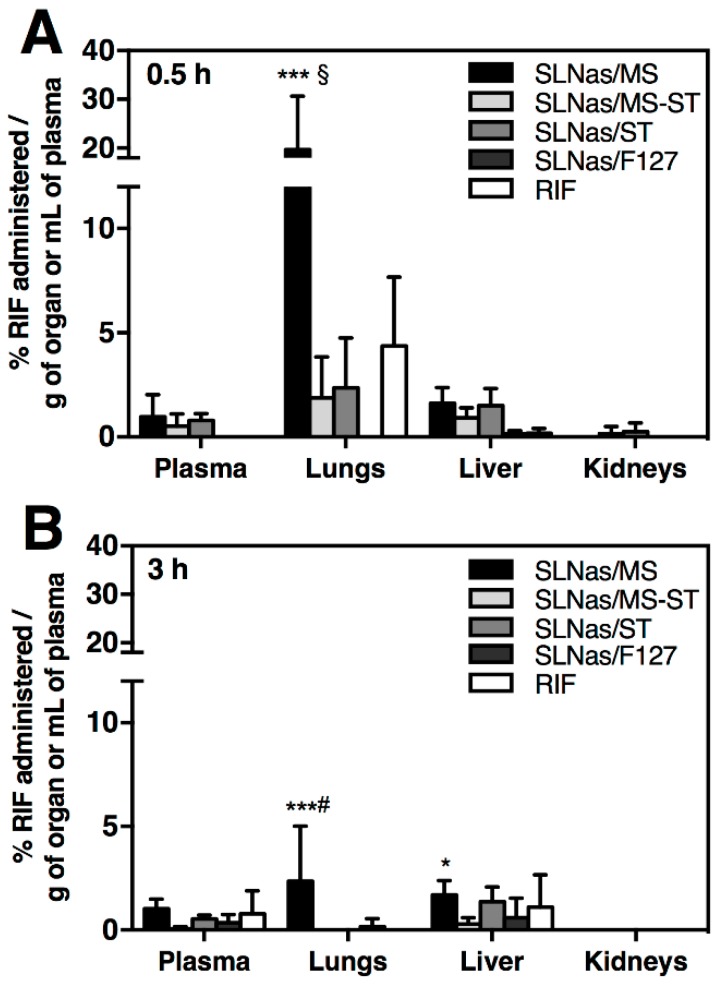
Quantification of RIF in plasma and tissue homogenates. Percentages of RIF dose per plasma volume or lung, liver, and kidneys weight (n = 4–6) at 0.5 h (**A**) and 3 h (**B**) after intratracheal administration of SLNas/MS, SLNas/MS-ST, SLNas/ST, SLNas/F127, and bare RIF. ***^§^
*p* < 0.001 vs. SLNas/MS-ST, SLNas/ST, SLNas/F127, and bare RIF. ***^#^
*p* < 0.001 vs. SLNas/MS-ST, SLNas/ST, SLNas/F127 and bare RIF. * *p* < 0.05 vs. SLNas/MS-ST.

**Figure 6 nanomaterials-10-00568-f006:**
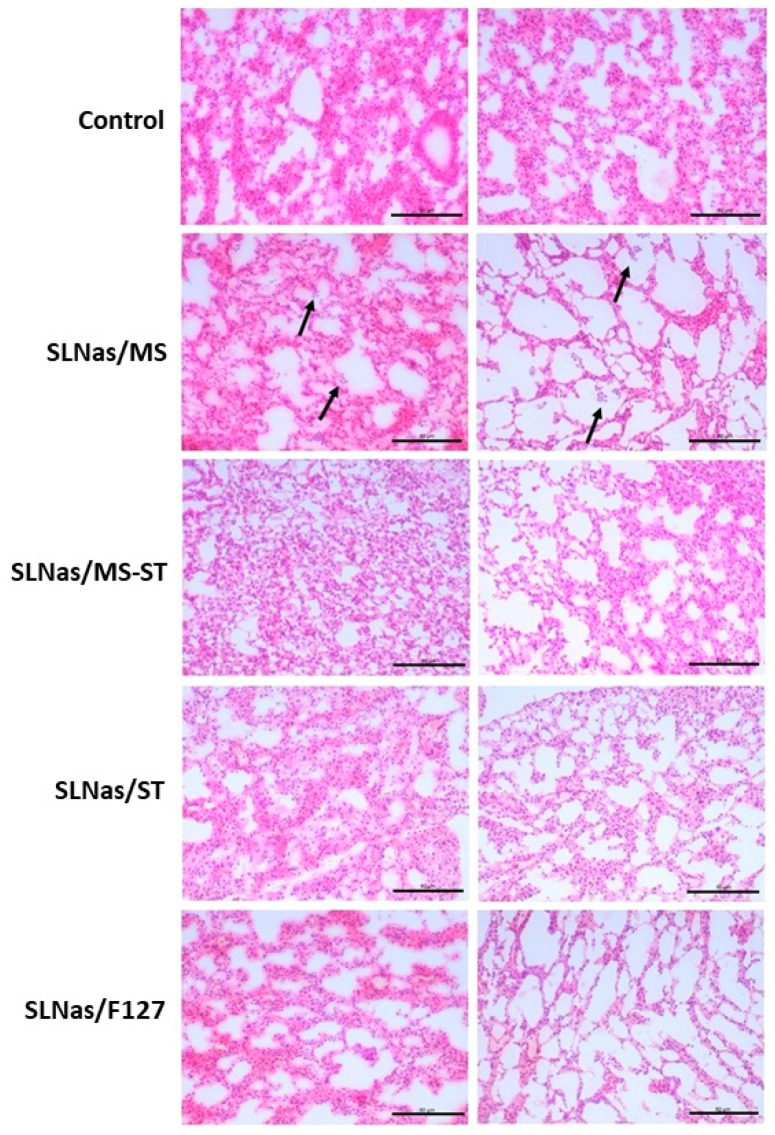
Histological sections of the medial (left panel) and inferior portion (right panel) of control mice lungs in comparison with lungs at 24 h post-exposure to SLNas/MS, SLNas/MS-ST, SLNas/ST, and SLNas/F127. Arrows indicate mild neutrophilic infiltration. Slides were stained using H&E. Magnification: 20×. Scale bars = 80 µm.

**Table 1 nanomaterials-10-00568-t001:** Size, PDI, Z-potential, and drug loading levels of the SLNas samples (mean values ± SD)

	SLNas/MS	SLNas/MS-ST	SLNas/ST	SLNas/F127
Size (nm)	559 ± 113	452 ± 92	520 ± 11	855 ± 97
PDI	0.68 ± 0.10	0.41 ± 0.02	0.56 ± 0.03	0.71 ± 0.06
Z-potential (mV)	−43.1 ± 1.6	−39.4 ± 1.9	−54.6 ± 2.2	−8.5 ± 0.3
Drug loading (%, w/w)	9.6 ± 0.2	10.3 ± 0.4	11.8 ± 0.4	8.10 ± 0.1

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
