# Peer review of "In Vivo Biodistribution of Respirable Solid Lipid Nanoparticles Surface-Decorated with a Mannose-Based Surfactant: A Promising Tool for Pulmonary Tuberculosis Treatment?"

_nanomaterials, 2020, doi:10.3390/nano10030568_

Round 1
Reviewer 1 Report
The current work related to the targeted uptake of solid lipid nanoparticles in an animal model is a good piece of work. The results documented are in-line with their hypothesis and it is convincing, yet some modifications could be considered before its publication.
Line 57-58: “Among these, lipid-based particles are composed of lipids generally recognized as safe and devoid of toxicity following pulmonary administration [11,12]” this statement conveys the lipid carriers are safe but this may not be true for all the lipids so generalization should be avoided.
Line 109: Is the structure of nanoparticles affected by freeze-drying? What is the lowest temperature used to freeze the samples?
Figure 1: The particles are looking very small in TEM imaging and the reported values in table 1 seem to be very high. Please check that and change with better quality images.
Figure 3: Before the administration of any formulation, there is a slight background coming from various regions of the mice. Could this be subtracted, thus you can see only the signal from the formulations?
Figure 4: please improve the image quality and minimize the background signals.
What is the therapeutic potential of these materials? Is it possible to show the activity of the various materials on the Mycobacterium tuberculosis-infected animal models?
Reviewer 2 Report
The work is interesting and well performed, The manuscript is well written. Therefore it is suitable for publication in this journal.
Author Response
We thank the Reviewer for his appreciation.
Reviewer 3 Report
Dear Editors,
It is my personal view that the MS entitled" In vivo biodistribution of respirable Solid Lipid Nanoparticles surface-decorated with a mannose-based surfactant: a promising tool for pulmonary tuberculosis treatment?" is a well written article with very useful information. It is ready to be published the final check is completed.
One question for authors. Just wondering if authors compare the Eosinophils?
Thank you very much!
Tom
Author Response
We thank the Reviewer for his appreciation. With regard to his question, in the BALF, the majority of the cells were macrophages and for this reason no comparisons have been made.
Regarding the histopathological analysis, in an acute inflammation, the neutrophils provide the second-line defense. They are the first cells to be recruited to sites of infection or injury. Usually, eosinophils are associated with chronic lung inflammatory states.